# Association between Sick Building Syndrome and Indoor Environmental Quality in Slovenian Hospitals: A Cross-Sectional Study

**DOI:** 10.3390/ijerph16173224

**Published:** 2019-09-03

**Authors:** Sedina Kalender Smajlović, Andreja Kukec, Mateja Dovjak

**Affiliations:** 1Angela Boškin Faculty of Health Care, Jesenice 4270, Slovenia; 2Faculty of Medicine, University of Ljubljana, Ljubljana 1000, Slovenia; 3Faculty of Civil and Geodetic Engineering, University of Ljubljana, Ljubljana 1000, Slovenia

**Keywords:** hospital ward, health risk factors, parameters, sick building syndrome, integral strategy, environmental health activities

## Abstract

Increased exposure times to various health risk factors and the vulnerability of building users might result in significantly higher prevalence rates of sick building syndrome (SBS) in a hospital setting compared to other indoor environments. The purpose of our study was to assess the association between SBS symptoms and measured environmental parameters at a Slovenian general hospital. A combination of a self-assessment study and field measurements was conducted in order to estimate the health risk factors for SBS symptoms among the users of a Slovenian general hospital. The Chi-square test was used to analyse the association between observed health and environmental parameters. The response rate was 67.5%. A total of 12.0% of healthcare workers at hospital wards reported at least six SBS symptoms, 19.0% reported 2–3 SBS symptoms. At the observed hospital wards, the most deviations were recorded for the level of lighting (83.3%), noise level (73.6%), and room temperature (55.3%). A statistically significant association was found between indoor environmental quality and skin-related SBS symptoms (χ^2^ = 0.009; *p* = 0.006). This information will be of great value in defining an integral strategy of environmental health activities aimed at healthier indoor environmental quality in hospitals.

## 1. Introduction

Environmental health risk factors and their parameters are defined as stressors which occupants are exposed to in their living and working environments and might potentially have harmful effects on their health, comfort and productivity [1,2,3]. On the basis of a comprehensive systematic literature review of 313 scientific articles published between 1974 and 2019 [3], key parameters and their possible interactions within every group of health risk factors were identified and classified into: biological (moulds, bacteria, microbial volatile organic compounds, house dust); chemical (construction and household products, formaldehyde, phthalates, man-made mineral fibres, volatile organic compounds, odours, environmental tobacco smoke, other indoor air pollutants); physical (environmental parameters of thermal comfort, parameters related to building ventilation, noise, vibrations, daylight, electromagnetic fields, ions, ergonomics, universal design); psychosocial (occupational stress, social status, loneliness, helplessness, work organization, communication, supervision); personal (gender, individual characteristics, health status); and other (location, geo-pathogenic zones; building characteristics, ownership, presence of insect, rodents, use of insecticide, disinfection, rat-killing products) [3].

Healthcare workers and healthcare associates in various areas of health services are exposed to numerous health risk factors connected to their occupational health and safety. The types of health risk factors are associated with the characteristics of the building, its heating, cooling, ventilation and air-conditioning (HVAC) systems, and the activities performed by the occupants/building users [4,5,6]. According to Gilligan et al. [7] and Ulutasdemir and Tanir [8], the health risk factors in health services are divided into biological, chemical, physical, ergonomic, and psychological, as well as workplace violence.

Most of the health risk factors and their parameters originate in the thermal, light, acoustic, olfactory and ergonomic environment; therefore, they are associated with indoor environmental quality (IEQ) issues [3]. Health risk factors in a hospital setting have previously been investigated by many authors. The first studies date back to the 1970s [9]; they focused on the associations between exposure to the chemical and biological health risk factors in hospitals. In the 1980s and 1990s, researchers mostly investigated physical health risk factors [10,11,12,13]. Conversely, more recent studies have focused not only on the identification of health risk factors in hospitals [14,15,16,17,18], but also on developing new approach strategies and innovative systems for their control and prevention [3,4,19].

The U.S. Environmental Protection Agency [20] describes sick building syndrome (SBS) as situations in which building occupants experience acute health - and comfort - related effects that appear to be linked to time spent in a building, without identified specific illness or cause. The complaints may be localized in a particular room or zone or may be widespread throughout the building. The characteristic symptoms of SBS that may occur singly or in combination with each other are headache, eye, nose, or throat irritation, dry cough, dry or itchy skin, dizziness and nausea, difficulty in concentrating, fatigue and sensitivity to odours. In contrast, the term Building Related Illness (BRI) is used when symptoms of diagnosable illness are identified and can be attributed directly to airborne building contaminants. 

Epidemiological studies around the world have emphasized that increased exposure times to various health risk factors and vulnerability of building occupants might result in significantly higher SBS prevalence in the hospital setting compared to other indoor environments. According to the latest findings, SBS prevalence range in hospitals is from 20.9% [21] to 86.4% [16]. The high prevalence of SBS symptoms was associated with the exposure to several health risk factors and their parameters. The most severe SBS symptoms were associated with the group of physical health risk factors, such as building dampness and the resulting moisture damage, lack of daylight, exposure to increased noise levels and decreased ventilation, and chemical health risk factors, especially related to poor indoor air quality (carbon dioxide—CO_2_, unpleasant odour, volatile organic compounds—VOC), as well as psychosocial factors (workload, decreased quality of working life) [16,22,23,24,25,26,27].

Extant research on SBS health risk factors in the hospital setting has mainly been conducted in different hospital wards among nurses [16,23,27], in neonatal intensive units among physicians and nurses [28], in medical centres among healthcare workers [25], in operating theatres and intensive care units among nurses and physicians [15], and in a paediatric dentistry clinic and a dental laboratory among dental professionals [29]. In addition, research [12,30] has included all hospital employees in public geriatric hospital units and a unit for non-geriatric patients. Research on SBS prevalence among patients has been scarcer: Edvardsson et al. [31] focused on SBS in patients at dermatology departments and the department of occupational and environmental medicine. Similarly, studies which systematically analyse the issue of SBS in hospitals and propose suggestions for improving the situation with a methodological approach are rare. However, it is crucial that in defining environmental health activities designed to control and prevent health hazards, an estimation of all risk factors and their parameters occurring in the hospital setting in relation to building users is made. 

The purpose of our study was to (1) estimate the prevalence of SBS symptoms in hospital wards, (2) to study the indoor environmental quality of selected hospital wards and compare them to the legally required and/or recommended values, and (3) to assess the association between SBS symptoms and indoor environmental quality. This information will be of great value in defining an integral strategy of environmental health activities aimed at healthier indoor environmental quality in hospitals.

## 2. Materials and Methods 

### 2.1. Study Design, Study Population/Sampling and Timeframe

The data were collected in the period February–April 2019 from a cross-sectional study conducted in Slovenia. A combination of a self-assessment study and field measurements were conducted. According to the small number of employees in the observed hospital, a convenience sample of healthcare workers and observed environmental parameters were used, in order to estimate the presence of SBS symptoms. Healthcare workers were employed at eight different wards of a Slovenian general hospital. 

### 2.2. Self-Assessment Questionnaire 

A validated, standardized self-assessment questionnaire was used to estimate the presence of SBS in the observed hospital. Participants were considered to suffer from SBS if they self-reported having one or more symptoms specified in the Indoor Climate Hospital/Health Care Establishment Questionnaire (MM 040 NA Hospital 2007) [32], which was used to collect information on environmental factors and typical SBS symptoms, including ocular, nasal, throat-related, skin-related, and non-specific building-related symptoms. 

Questionnaire (MM 040 NA Hospital 2007) included 16 assumptions selected according the purpose of our study. Assumptions collected an information on the presence of SBS symptoms over a period of the last three months. The presence of SBS symptoms was assessed with several questions and included: itching, burning or irritation of the eyes; irritated, stuffy or runny nose; hoarse, dry throat, cough; dry or flushed facial skin; scaling/itching scalp or ears; dry, itching, red skin on hands; as well as tiredness, fatigue, headache, difficulties concentrating, suffering from stress, anger and dizziness. 

SBS symptoms were assessed with the question: “Over the last three months, have you had any of the following symptoms?” Possible answers were: 1—yes, often, weekly; 2—yes, sometimes, 3— no, never. 

First, we “counted” the symptoms that were self-reported by each respondent: “Yes, often (weekly), over the last three months. Therefore, the overall number of SBS symptoms that were self-reported by one respondent was in the range of 0 (minimal) to 16 (maximal). With the formulated variable, the average number of symptoms per hospital ward can be compared. In order to compare the percentage of occurrence in different hospital wards, the symptoms were grouped in four groups: 0–1 symptom; 2–3 symptoms, 4–5 symptoms and 6 symptoms or more. Additionally, the symptoms were classified into five categories: nasal symptoms; ocular symptoms; throat-related symptoms; skin-related symptoms and general symptoms.

As a different number of indicators are combined into each element, we further introduced a dichotomous distribution (health risk factor was present/was not present). The risk factor was present if the answer “Yes, often (weekly), over the period of the last three months” appeared in at least one of the indicators, which falls into the group of certain SBS symptoms.

### 2.3. Environmental Parameters

In situ measurements at selected hospital wards were conducted using calibrated instruments. Environmental measurements included parameters related to IEQ: air quality (indicator: carbon dioxide (CO_2_) (ppm)), horizontal illumination (lx), indoor relative humidity level (%), indoor air temperature (°C), and noise level (dB(A)). Indoor air temperature, the indoor relative humidity level, and the noise level were measured with a Voltcraft UM5/1 100 (Conrad Electronic SE, Hirschau, Germany), CO_2_ concentrations were measured using a CO_2_ meter (CM-100, Voltcraft, Conrad Electronic SE, Hirschau, Germany), and horizontal illumination at working height was measured using an LED Light Meter LT40 (Taiwan). All levels were monitored in the morning when most healthcare workers were present at hospital wards. A total of 360 consecutive measurements (at two measurement locations in each ward) were conducted at the following wards/units: the surgical intensive care unit, which is part of the anaesthesiology and reanimation services; the emergency department; the nursing ward, which is part of the non-acute physical rehabilitation services; the joint medical departments; the department for medicinal product supplies; and the surgical admissions unit. Measurements were taken from 07:00 to 11:30 and entered into a checklist at the following wards/units: (1) anaesthesiology and reanimation services (measurement locations: A, working counter of a nurses station at the surgical intensive care unit; B, working counter for preparation of medications at the surgical intensive care unit); (2) the emergency department (measurement locations: C, working counter at the observation unit; D, overbed table as part of a hospital bed unit); (3) non-acute physical rehabilitation services (measurement locations: E, working counter of an intervention area at the nursing ward; F, working counter in a nurses’ room at the nursing ward); (4) joint medical departments (measurement locations: G, working counter for the analysis of laboratory blood testing; H, working counter for the analysis of laboratory biochemical tests); (5) department for medicinal product supplies (measurement locations: I, working counter for the preparation of narcotic drugs; J, working counter for issuing pharmacy materials), and at the (6) surgical admissions unit (measurement locations: K, working counter in the external admissions office for patients; L, working counter of the internal admissions office for patients).

In the data that was entered into the checklist, *0* signified no deviations from recommended values and *1* signified that measurements deviated from recommended values. The benchmarks were legally required and/or recommended values. In measurements of carbon dioxide (CO_2_), deviations constituted any value over 1000 ppm, recommended by The American Society of Heating, Refrigerating and Air-Conditioning Engineers [33]. For indoor air temperature, deviations constituted values under 20 °C and over 22 °C during the heating season, required by national rules [34]. For horizontal illumination, deviations constituted any value that differed by less than 500 lx for general lighting, and by less than 1000 lx for lighting used during medical examinations of patients and during healthcare provision in hospital spaces. For measurement locations where medical and nursing procedures were documented, the benchmark used was the reading light standard of less than 300 lx, recommended by EN 12464-1:2011 [35]. Both natural and artificial light were taken into account in illumination measurements. For the level of indoor relative humidity, deviation constituted any value under 30% and over 60% at room temperature of 20–25 °C, recommended by ANSI/ASHRAE/ASHE Standards 170-2008 [36]. Finally, deviations in noise level constituted a noise level of over 55 dB(A) in hospital wards, required by national rules [37]. 

### 2.4. Observed Outcome

SBS symptoms were assessed with two questions: (1) “Over the last three months, have you had any of the following symptoms (fatigue, headache, etc.)?” Possible answers were: 1—yes, often, 2—yes, sometimes, 3—no, never. (2) “Do you believe that this was due to your work environment?” The answers were: 1—yes, 2—no, 3—I don’t know.

### 2.5. Other Risk Factors

Other risk factors included gender, age group, work environment, work conditions, previous/present diseases, and work environment in general. 

### 2.6. Methods of Analysis

The distributions of observed outcomes, environmental parameters and other risk factors were described with descriptive statistics. The Chi-square test was used to test the association between the presence of SBS symptoms and measured parameters of indoor environmental quality in the selected hospital environment. The level of statistical significance was set at *p* ≤ 0.05 for all statistical tests. The SPSS statistical software for Windows (Version 21.0; IBM Corp.; Armonk, NY, USA) (License: University of Ljubljana, Slovenia) was used as an analysis tool.

### 2.7. Ethical Consideration

The protocol of our study was approved by The National Medical Ethics Committee of the Republic of Slovenia in August 2018; Code Number: 0120-135/2018/10.

## 3. Results

### General Characteristics of the Participants

The reliability of questions of SBS symptoms of the MM040 was calculated by Cronbach’s alpha coefficient. The coefficient by healthcare workers was 0.825 and healthcare associates was 0.874. Table 1 shows the sample characteristic. A total of 258 healthcare workers were included in our study. The response rate was 67.5%. Of the respondents, 82.6% were female and 17.4% male. They provided healthcare services at different hospital wards (most worked at surgical wards (26.4%), followed by joint medical departments (21.3%), internal medicine wards (14.9%), the emergency department (10.5%), anaesthesiology and reanimation services (10.1%), the department of gynaecology and obstetrics (8.1%), the paediatric ward (5.0%) and, finally, the non-acute physical rehabilitation services (4.3%)).

The mean age of study respondents was 37 years, their mean period of employment in the hospital setting and in their current position was 15.1 years and 9.9 years, respectively. Most healthcare worker respondents (60.9%) held a higher education degree or higher. Nursing professionals constituted the majority of healthcare worker respondents (75.6%), while physicians made up the smallest respondent group (8.9%). In terms of their smoker status, 60.1% of respondents reported being non-smokers. A total of 43.4% healthcare workers reported working overtime; of these, 67.1% also reported working shifts.

Table 2 shows the prevalence of self-perceived SBS symptoms in healthcare workers according to hospital wards. Significant association was found to exist between the number of self-perceived SBS symptoms among healthcare workers and the hospital ward type (χ^2^ = 37,698; *p* = 0.014). A total of 12.0% of hospital healthcare workers reported experiencing six or more SBS symptoms, and 19.0% reported experiencing two to three SBS symptoms. A significant association was established between nasal symptoms and the type of hospital ward, with most nasal symptoms being reported by healthcare workers at the paediatric ward (23.1%), and fewest at the anaesthesiology and reanimation services and the department of gynaecology and obstetrics (0%). In addition, a significant association was also found to exist between skin-related SBS symptoms and hospital ward type, with healthcare workers at the surgical ward reporting the highest prevalence (36.8%) and those working at the non-acute physical rehabilitation services reporting the lowest (0%). Finally, a significant association was also established between general SBS symptoms and the type of hospital ward the highest prevalence was again established for the surgical ward (67.6%) and the lowest for the anaesthesiology and reanimation services. 

Table 3 shows the obtained measurements for the indoor environmental quality parameters in the selected hospital environment. A significant association was established between the number of persons present in the space and measurement locations at the selected hospital wards. Most people were present at both measurement locations at the emergency department with a floor space of 800 m^2^ (measurement locations C and D; mean 7.7; approximately 0.009 persons/m^2^), and the fewest at the surgical admissions unit (external admission office) with a floor space of 11.25 m^2^ (measurement point K; mean 2.4; approximately 0.213 persons/m^2^). 

For air temperature, a significant association was established among measurement locations of the selected hospital wards. Air temperature was highest at the anaesthesiology and reanimation services (surgical intensive care unit) (23.9 °C) and lowest at the non-acute physical rehabilitation services (19.1 °C). A significant association was also found to exist between measurement locations of the selected hospital wards for the level of relative humidity. The highest level of relative humidity was measured at the non-acute physical rehabilitation services (49.9%), while the lowest was obtained at the emergency department (18.0%).

Similarly, a significant association among measurement locations of the selected hospital wards was established for the level of lighting (χ^2^ = 218.758; *p* = 0.000). The highest mean level of lighting was established at the department for medicinal product supplies (the pharmacy) (mean 534.4 lx), and the lowest at the surgical admissions unit (mean 120.2 lx).

Differences among the measurement locations at the selected hospital wards were also established for CO_2_ concentrations. The highest mean concentration of CO_2_ was measured at the surgical admissions unit (mean 1106.4 ppm), and the lowest at the emergency department (mean 594.9 ppm).

Finally, differences among the measurement locations at the selected hospital wards were established for the noise level. The highest mean noise level was recorded at the surgical admissions unit (mean 71.1 dB(A)), and the lowest at the department for medicinal product supplies (mean 50.6 dB(A)).

Table 4 shows the deviations of parameters according to measurement locations of the selected hospital wards. A comparison with values which are required by law or recommended revealed that, at all hospital wards, the parameter for the level of lighting deviated most (in 83.3%), followed by noise level (in 73.6%). The lowest deviations (in 21.7%) from legally required or recommended values at all the selected hospital wards were established for the level of relative humidity. 

At the anaesthesiology and reanimation services (measurement locations A and B), the level of lighting deviated most from the legally required and/or recommended values (100% at measurement locations A and B), followed by deviations in the noise level (measurement location B: 83.3%).

At the emergency department, the level of lighting and air temperature deviated in 100% of cases (measurement location D); a high deviation was also established for the noise level (measurement location D: 90.0%).

At the non-acute physical rehabilitation services, the greatest deviations were recorded for the level of lighting (measurement location E: 86.7%), followed by noise level (measurement location F: 80.0%).

Similarly, the level of lighting was also the parameter which deviated most from the legally required and/or recommended values at the joint medical departments (in 100%), followed by noise level (measurement location H. 93.3%; measurement location G: 100%).

At the department for medicinal product supplies, no deviations were found to exist for the level of relative humidity and the level of CO_2_ (both 0%); this department also had the fewest deviations in noise level (measurement location I.: 26.7%; measurement location J: 33.3%) and the level of lighting compared to other selected hospital wards (measurement location I: 10.0%; measurement location J: 36.7%). 

At the surgical admissions unit, the parameters which revealed deviations were level of lighting (in 100%), CO_2_ concentrations (measurement location K: 43.3%; measurement location L: 96.7%)—here, the mean CO_2_ concentration was highest for both measurement locations compared to other results (measurement location K: mean 1004.9 ppm; measurement location L: mean 1106.4 ppm)—and noise level (in 90% and more for both measurement locations).

Air temperature was the parameter which showed the greatest deviations from legally required and/or recommended values at the emergency department (the overbed table measurement location in 100%), and lowest deviations at the surgical admissions unit (working counter in the external admissions office for patients in 16.7%). The level of relative humidity deviated most at the emergency department, at both measurement locations (in 66.7%), while no deviations (0%) were recorded at the department for medicinal product supplies (both measurement locations), the anaesthesiology and reanimation services (measurement location B), and the surgical admissions unit (measurement location L). Conversely, deviations were established in 100% of measurements for the level of lighting at the anaesthesiology and reanimation services, the emergency department, the joint medical departments, and the surgical admissions unit (all measurement locations). The fewest deviations (in 10%) in the level of lighting were established at the department for medicinal product supplies (measurement location I). The CO_2_ concentrations were found to deviate most from required/recommended values at the surgical admissions unit (in 96.7%), while no deviations (0%) were recorded at the emergency department (measurement location C), the joint medical departments, and the department for medicinal product supplies. Finally, noise levels were highest at the joint medical departments (measurement location G; 100% of deviations), and lowest at the department for medicinal product supplies (measurement location I; 26.7% deviations).

No statistically significant differences (χ^2^ = 2.159; *p* = 0.093) were established between the number of SBS symptoms among healthcare workers employed at the selected hospital wards and the obtained mean values for air temperature, level of relative humidity, level of lighting, CO_2_ concentrations and noise level.

A significant association was found to exist between IEQ and skin-related SBS symptoms (χ^2^ = 0.009; *p* = 0.006). Conversely, no significant association was established between IEQ and nasal SBS symptoms (χ^2^ = 0.017; *p* = 0.567), ocular SBS symptoms (χ^2^ = 0.339; *p* = 0.170), throat-related SBS symptoms (χ^2^ = 1.000; *p* = 0.600), and general SBS symptoms (χ^2^ = 0.249; *p*= 0.139).

## 4. Discussion

There is nowadays a relatively large body of epidemiological research evidence available on the possible effects of an unhealthy indoor environment in hospitals on the health, comfort and productivity of hospital users. As the primary health outcome in hospital settings, SBS became the focus of research in the 1990s and continues to be researched to this day by many authors worldwide [12,13,30,38,39,40] who have emphasized the issue of exposure to health risk factors in hospitals which could result in a high SBS prevalence. They have stressed the urgency of implementing activities aimed at solving this issue. With a case study of a Slovenian general hospital, we estimated the health risk factors for SBS and looked for possible associations with the aim of defining an integral strategy of environmental health activities geared toward higher indoor environmental quality in hospitals.

### 4.1. Number of SBS Symptoms and Hospital Ward Type

The conducted cross-sectional study on 258 healthcare workers employed at a Slovenian general hospital revealed significant association between the number of self-perceived SBS symptoms and hospital ward type (χ^2^ = 37.698; *p* = 0.014). The greatest percentage of hospital healthcare workers (12.0%) reported experiencing six SBS symptoms, followed by those experiencing two to three SBS symptoms (19.0%). There is little other research evidence on the association between the number of SBS symptoms and hospital ward type, so no comprehensive comparison can be made. In their research among 126 hospital workers at a medical centre in Taiwan, Chang et al. [25] found that hospital workers suffered from at least one SBS symptom. Surgical units are environments with a certain amount of risk, especially due to the possible exposure to various chemical, biologic or physical hazards [41]. Delclos et al. [42] even state that the influence of occupational exposures on asthma in healthcare professionals is not trivial, as they found an increased risk of work-related asthma among healthcare workers. Nordström et al. [13] investigated general SBS symptoms in new buildings and buildings with a high ventilation flow among 225 female hospital workers. General symptoms, for example headache and fatigue, were related to smoking, asthma or hay fever, work dissatisfaction and static electricity. Facial skin irritation was related to a lack of control of the work conditions; it was more frequent in new buildings and buildings with a high ventilation flow and ventilation noise. Conversely to our study, where one general hospital was investigated, Kelland [10] looked at two London teaching hospitals: one was a modern building with artificial ventilation, the other was older and relied on natural ventilation. Staff were selected at random over a 2 month period and administered a questionnaire. Both staff groups at the first hospital experienced a higher symptom rate compared to those at the older hospital; this was related to a low perceived quality of the working environment, characterized by the perception of dryness, heat and low environmental control. An increased symptom rate could have possibly resulted from poor air-conditioning at the hospital due to cost-cutting measures imposed by hospital managers.

### 4.2. Classification of SBS Symptoms and Hospital Ward Type 

In our study, SBS symptoms were classified according to the ECA [43]; among them, the most frequent complaints were nasal symptoms, non-specific symptoms, and skin-related symptoms. A significant association was found to exist between SBS symptoms and hospital ward type, with the surgical ward and the paediatric ward being the most problematic. A significant association existed between skin-related SBS symptoms and hospital ward type (χ^2^ = 16.657; *p* = 0.020), with healthcare workers employed at the surgical ward reporting the highest prevalence of skin-related symptoms (36.8%). In addition, a significant association was established between general SBS symptoms and hospital ward type (χ^2^ = 14.999; *p* = 0.036), with the highest prevalence again being reported by those working at the surgical ward (67.6%). A significant association also existed between nasal SBS symptoms and hospital ward type (χ^2^ = 11.407; *p* = 0.012), with the highest prevalence of nasal SBS symptoms reported by those working at the paediatric ward (23.1%). Although some other authors do not classify SBS symptoms according to the ECA [43], their findings were similar. For example, Nordström et al. [12] showed that in eight hospital units in Sweden (*n* = 225 female hospital workers), the prevalence of SBS differed from one unit to another. The mean values of weekly complaints of fatigue, eye irritation, and dry facial skin were 30.0%, 23.0% and 34.0%, respectively. In a study by Chang et al. [25], the most frequently reported symptoms in a medical centre in Taiwan (*n* = 126 hospital workers) were nasal symptoms (66.0%), followed by ocular symptoms (53.0%), fatigue (30.0%), headache (19.0%), and facial dryness (33.0%). A cross-sectional study conducted in neonatal intensive care units on a sample of physicians and nurses [28] found that approximately 60.0% of healthcare workers suffered from SBS in the form of fatigue (83.0%) and headaches (76.0%). What is more, in their research among 265 nurses working in teaching hospitals of Shahid Sadoughi University of Medical Sciences in Yazd, Iran, Vafaeenasab et al. [16] established that the prevalence of SBS was 86.4%, and that the most frequent complaints reported by nurses included headache, fatigue, and dryness of the hands.

### 4.3. Inappropriate Indoor Environmental Quality Parameters

In our study, we compared the obtained indoor environment parameters with the legally required and/or recommended values. Overall, we established the following for all measurement locations (A–L): the level of lighting was too low (in 83.3%), noise level was too high (in 73.6%), and air temperature was inappropriate (in 55.3%). The inappropriate self-perceived parameters reported by healthcare workers were: poor air quality (51.6%), level of humidity (49.6%), and air temperature (41.1%); healthcare worker respondents seemed to be least bothered by the level of lighting (34.1%) and noise level (22.5%). All inappropriate parameters—both the measured and the self-perceived ones—constitute a possible physical risk factor. We could find no other research assessing measurement locations in hospital wards in relation to SBS in a similar manner. Nevertheless, several studies which investigated aspects of IEQ and SBS in a hospital setting established similar risk factors, including level of lighting, noise level, and air quality. Italian researchers [15] measured the subjective emotional discomfort (stress) among nurses and anaesthesiologists (*n* = 134) working in operating theatres and intensive care units, and its association with environmental discomfort factors, especially the level of lighting. They found that, upon comparing different types of exposure with horizontal illumination values and the degree of stress reported, the percentage of high stress was reduced with an increase in the exposure to illumination, although the finding was not statistically significant. The relationship between psychosocial factors and the perception of the work environment was emphasized in the several Italian studies [44,45]. Workplace violence and occupational stress, as important psychosocial risk factors for SBS, were researched by Magnavita [46]. A result of a 6 year follow-up study on healthcare workers (*n* = 698) showed that job strain and lack of social support were predictors of the occurrence of non-physical aggression during the ensuing year. Health care workers who experienced workplace violence reported high strain and low support at work in the following year. The experience of non-physical violence and a prolonged state of strain and social isolation were significant predictors of psychological problems and bad health at follow-up. Additionally, Ferry et al. [47] conducted a cross-sectional study in 17 wards of a general hospital and a residential facility of a northern Italian city and involved 213 nurses working in rotating night and day shifts. The authors highlighted that shift work with nights, as compared with day work only, was associated with risk factors predisposing nurses to poorer health conditions and lower job satisfaction.

In terms of excessive noise levels, a cross-sectional study (*n* = 177) by Arikan et al. [21] established that the risk for SBS was 1.2 times higher with an increase in noise level measurements, and 2.1 times higher with an increase in CO_2_ concentration measurements. Lack of a sense of airflow, unpleasant odours in the workplace (*p* < 0.05), and the amount of workload (*p* < 0.001) were all found to be related to SBS. In Iran, research by Keyvani et al. [26] on 41 healthcare workers revealed a high prevalence of SBS among staff to be related to such factors as unpleasant odours, noise level, and a low level of lighting. Similar conclusions with respect to noise level were also reached by other researchers. For example, Rashid and Zimring [48] found that the hospital environment was characterized by a high level of noise and that the noise level tended to exceed the World Health Organization (WHO) guidelines. Through an epidemiological and environmental assessment, Niven et al. [14] reported consistent findings for noise variables and determined that low-frequency noise was directly associated with such symptoms as stuffy nose, itchy eyes, and dryness of the skin. Moreover, Nordström et al. [13] found that eye irritation was related to work stress and self-perceived exposure to static electricity; it was more common in buildings with a high ventilation flow and a high noise level (55 dB(A)) from the ventilation system. In terms of air quality, other authors [38] found that CO_2_ concentration did not associate with SBS. Also, in Turkey, researchers [27] measured the indoor air quality at eight different locations at a hospital; their measurements included carbon dioxide concentration, carbon monoxide concentration, temperature, level of humidity, and microbiological matter. The highest CO_2_ concentration was obtained at the paediatric clinic. Poor interior air quality results are the most significant cross-sectional data.

### 4.4. Methodological Approach

It is important to remember that any deviation from legally required or recommended values of IEQ parameters could result in SBS symptoms. Therefore, a multi-level, comprehensive approach is required (entailing a combination of methodological approaches for exposure assessment: questionnaire, in-situ measurements). We propose the use of a methodological approach aimed at preventing and controlling SBS, based on the findings of SBS prevalence; next, the identification of health factors by measuring selected IEQ parameters in order to reveal problematic areas (deviations from the legally required/recommended values), and the implementation of a self-assessment (users’ answers to questions on inappropriate IEQ parameters). Statistical analyses of associations between self-assessment results, risk factors, and environmental measures also represent highly valuable information. The method enables fast identification of problematic areas, and consequently fast implementation of step-by-step measures and activities related to the renovation of buildings and systems, and to raising users’ awareness of activities aimed at planning healthy buildings and hospitals [3]. In order to improve the quality of the indoor environment, the building design has to be adapted to climate characteristics and changes [49]. As has previously been stated by Cornelio et al. [50], variables should also be included from the psychosocial and gender points of view in designing and targeting preventive strategies, and for a comprehensive understanding of health outcomes in the working population, as there continues to be growing evidence of health outcomes related to environmental factors. We recommend that further measurements of selected parameters be conducted in the hospital environment. Health care workers are exposed to conditions that might increase the risk for a variety of SBS symptoms. Although all measured parameters of IEQ in the hospital exceeded the recommended limits, long-term exposures should be considered for those suffering from SBS symptoms. We recommend the implementation of a uniform action model for resolving indoor air problems in the hospital setting, consisting of representatives from occupational health, occupational safety, and healthcare workers from infection control. A method of evaluation by steps including a subjective assessment and technical measurements proposed by Chirico and Rully [51] should be used in the hospital environment. More focus should be placed on IEQ at hospitals wards—improvements to the work environment may be the most cost-effective way of reducing the burden of indoor exposure.

### 4.5. Strengths and Limitations 

The present study is the first in Slovenia to estimate SBS-related health risk factors and their parameters in relation to hospital users, and to assess the association between identified health risk factors, SBS symptoms and measured environmental parameters at a Slovenian general hospital. Based on the identified association between environmental parameters and SBS symptoms, we recommend environmental health activities aimed at a healthier hospital environment. 

The strengths of the present study are that we looked not only at subjective SBS symptoms, but also implemented measurements of parameters and identified health risk factors among hospital healthcare workers—the most numerous and most recognized population group in a general hospital setting. Collecting data on subjective SBS symptoms and health risk factors was a fast way to obtain data on SBS prevalence among the observed population—healthcare workers. The method can be conducted with minimum cost and be put into practice relatively quickly. In order to obtain final, reliable results, we recommend conducting a further in-depth analysis together with designing and testing a methodological approach.

A limitation of our study is that we measured parameters over a very limited time period. In the future, we plan on measuring parameters over a longer period of time, including the winter season. In addition, measurements were not conducted at all wards and patients in hospital rooms could not always be included due to ethical requirements of the study. Additionally, researcher Magnavita [45,46] highlighted an important relationship between psychosocial factors and the perception of the work environment. Therefore, all those relationships should be analysed in our future work, and MM040/IAQ questionnaire proposed by Magnavita [44] should be applied. Finally, an additional limitation is the small sample size of physicians in the self-assessment method.

Despite existing legally required and/or recommended values of selected IEQ parameters, we recommend a multi-level analysis in the methodological approach, as designing an integral, targeted approach will enable a quick assessment to be made and targeted measures to be adopted when the first subjective SBS symptoms arise, and in all further ways of addressing SBS.

The results presented in this paper should provide guidance in planning future evidence-based public health activities:it is crucial to raise the awareness of SBS among healthcare workers, and to make hospital managers aware of the importance of maintaining and promoting health in work environments;our study represents a platform for the identification of health risk factors and, as such, serves as a starting point for further activities;further activities include: implementation of cross-sectional analyses using a valid instrument in order to obtain data on SBS prevalence; identification of risk factors in the working environment; implementation of initial and periodic measurements of IEQ indicators and their comparison to the legally required and/or recommended values; identification of high-risk work spaces; identification and implementation of necessary professional measures; obtaining statistical data; and integration of the knowledge obtained from the fields of public health, planning and construction of hospitals, and healthy working environments. Finally, as it was highlighted by Chirico and Magnavita [52]: “every technical, procedural, or organizational measures could be ineffective without taking into account the results provided by health surveillance”.

## 5. Conclusions

Our study established:a higher prevalence of SBS symptoms among healthcare workers; however, no significant association was found to exist between the number of SBS symptoms in healthcare workers employed at the selected hospital wards and the measured mean values for air temperature, level of relative humidity, level of lighting, air quality (CO_2_ concentration), and noise level;a significant association between IEQ and skin-related SBS symptoms, while no significant associations were established between IEQ and nasal symptoms, ocular symptoms, throat-related symptoms, and general SBS symptoms;differences in environmental parameters at the selected hospital wards, based on actual measurements, and their comparison with the legally required and/or recommended values;that healthcare workers self-perceive the main risk factors to be poor air quality, inappropriate level of relative humidity, and inappropriate room temperature, while the least relevant risk factors were level of lighting and noise level. The mentioned risk factors could result in the prevalence of SBS; these findings could be of great value in defining an integral strategy of environmental health activities aimed at healthier IEQ in hospitals.

## Figures and Tables

**Table 1 ijerph-16-03224-t001:** General characteristics of healthcare workers at a Slovenian general hospital (*n* = 258).

Total (*n* = 258)				D. of Surgery (*n* = 68, 26.4%)		D. of Anaesthesiology and Reanimation (*n* = 26,10.1%)		D. of Gynaecology and Obstetrics (*n* = 21, 8.1%)		D. of Internal Medicine (*n* = 37, 14.9%)		D. of Paediatrics (*n* = 13, 5%)		Non-Acute Physical Rehabilitation Services (*n* = 11, 4.3%)		Emergency Department (*n* = 27; 10.5%)		Employing other Healthcare Workers (*n* = 55, 21.3%)	
male	f/%	45	17.4	15	22.1	6	23.1	0	0	7	18.9	0	0	1	9.1	4	14.8	12	21.8
female	f/%	213	82.6	53	77.9	20	76.9	21	100.0	30	81.1	13	100.0	10	90.9	23	85.2	43	78.2
age	M/SD	37.0	10.4	33.2	9.6	38.8	9.6	42.6	12.1	35.0	8.8	38.9	9.4	39.0	9.8	34.4	9.0	40.6	11.0
total period of employment	M/SD	15.1	11.1	11.5	9.5	17.2	10.0	21.3	13.5	12.8	10.0	18.7	9.6	18.9	10.3	12.1	10.7	17.7	12.0
period of employment at current position	M/SD	9.9	9.7	7.7	8.5	9.3	8.6	17.8	13.9	8.7	8.5	15.0	9.7	6.9	6.3	6.1	5.8	12.1	10.5
elementary–non-university higher education	f/%	101	39.1	38	55.9	3	11.5	13	61.9	9	24.3	10	76.9	8	72.7	3	11.1	17	30.9
university-level education or higher	f/%	157	60.9	30	44.1	23	88.5	8	38.1	28	75.7	3	23.1	3	27.3	24	88.9	38	69.1
physicians	f/%	23	8.9	8	11.8	3	11.5	1	4.8	2	5.4	0	0.0	0	0.0	4	14.8	5	9.1
nursing and midwifery professionals	f/%	195	75.6	59	86.8	23	88.5	20	95.2	35	94.6	13	100.0	10	90.9	23	85.2	12	21.8
other healthcare employees	f/%	40	15.5	1	1.5	0	0.0	0	0.0	0	0.0	0	0.0	1	9.1	0	0.0	38	69.1
non-smokers	f/%	155	60.1	36	52.9	14	53.8	13	61.9	23	62.2	9	69.2	4	36.4	17	63.0	39	70.9
overtime work	f/%	112	43.4	31	45.6	12	46.2	10	47.6	25	67.6	6	46.2	7	63.6	12	44.4	9	16.4
shift work	f/%	173	67.1	51	75.0	16	61.5	19	90.5	24	64.9	12	92.3	7	63.6	24	88.9	20	36.4

Note: D, Department, *n*, sample size; f, frequency; %, percentage; M, Mean; SD, Standard Deviation.

**Table 2 ijerph-16-03224-t002:** Prevalence of self-perceived sick building syndrome (SBS) symptoms among healthcare workers.

Total (*n* = 258)				Surgical Ward (*n* = 68)		Anaesthesiology and Reanimation Services (*n* = 26)		Gynaecology and Obstetrics (*n* = 21)		Internal Medicine (*n* = 37)		Paediatric Ward (*n* = 13)		Non-Acute Physical Rehabilitation Services (*n* = 11)		Emergency Department (*n* = 27)		Joint Medical Departments (*n* = 55)		Mann–Whitney U/*Chi*-square	df	*p*
Number of symptoms (min 0/max 16)	Mean and SD	2.03	2.55	3.07	3.05	1.27	1.99	1.38	2.06	2.38	2.81	2.15	2.67	1.73	2.05	1.63	2.04	1.33	1.98	12.837	6	0.045 *
Number of symptoms	0–1 symptom (f in %)	149	57.8	26	38.2	20	76.9	16	76.2	19	51.4	8	61.5	5	45.5	16	59.3	39	70.9	37.698	21	0.014 *
	2–3 symptoms (f and %)	49	19.0	15	22.1	2	7.7	2	9.5	7	18.9	2	15.4	5	45.5	5	18.5	11	20.0			
	4–5 symptoms (f and %)	29	11.2	13	19.1	3	11.5	2	9.5	4	10.8	1	7.7	0	0.0	4	14.8	2	3.6			
	6 symptoms or more (f and %)	31	12.0	14	20.6	1	3.8	1	4.8	7	18.9	2	15.4	1	9.1	2	7.4	3	5.5			
Nasal symptoms	Yes, frequently (f and %)	21	8.1	6	8.8	0	0.0	0	0.0	4	10.8	3	23.1	1	9.1	3	11.1	4	7.3	11.407	7.0	0.012 *
Ocular symptoms	Yes, frequently (f and %)	26	10.1	10	14.7	2	7.7	1	4.8	3	8.1	2	15.4	1	9.1	4	14.8	3	5.5	5.062	7.0	0.652
Throat-related symptoms	Yes, frequently (f and %)	16	6.2	5	7.4	0	0.0	0	0.0	4	10.8	1	7.7	1	9.1	2	7.4	3	5.5	7.589	7.0	0.370
Skin-related symptoms	Yes, frequently (f and %)	58	22.5	25	36.8	6	23.1	5	23.8	7	18.9	3	23.1	0	0.0	5	18.5	7	12.7	16.657	7.0	0.020 *
General symptoms	Yes, frequently (f and %)	134	51.9	46	67.6	9	34.6	10	47.6	20	54.1	8	61.5	7	63.6	11	40.7	23	41.8	14.999	7.0	0.036 *

Note: *n*, sample size; f, frequency; %, percentage; SD, Standard Deviation; df, degrees of freedom; * *p* < 0.05; (Mann–Whitney test/Chi-square test).

**Table 3 ijerph-16-03224-t003:** Obtained indoor environmental quality parameters (min, max, median, mean) for the selected measurement locations of hospital wards (A–L).

Measured Parameters of Indoor Environmental Quality (Min, Max, Median, Mean) (*n* = 360)			Hospital Ward and Letter Signifying Measurement Location at the Ward												Differences Obtained with KW Test/Chi-Square Test		
			Anaesthesiology and Reanimation Services		Emergency Department		Non-Acute physical Rehabilitation Services		Joint Medical Departments		Department for Medicinal Product Supplies		Surgical Admissions Unit		Chi-square	df	*p*
			A (*n* = 30)	B (*n* = 30)	C (*n* = 30)	D (*n* = 30)	E (*n* = 30)	F (*n* = 30)	G (*n* = 30)	H (*n* = 30)	I (*n* = 30)	J (*n* = 30)	K (*n* = 30)	L (*n* = 30)			
Number of people in the space	Min	2.0	4.0	4.0	7.0	7.0	5.0	5.0	6.0	6.0	5.0	5.0	2.0	5.0	303.427	11	<0.001 *
	Max	8.0	5.0	5.0	8.0	8.0	6.0	6.0	7.0	7.0	5.0	5.0	6.0	7.0			
	Median	6.0	5.0	5.0	8.0	8.0	6.0	6.0	6.0	6.0	5.0	5.0	2.0	6.0			
	Mean	5.6	4.7	4.7	7.7	7.7	5.7	5.7	6.3	6.3	5.0	5.0	2.4	6.0			
Air temperature (°C)	Min	19.1	20.7	21.2	21.0	22.1	19.1	20.6	21.4	20.6	20.7	21.4	20.4	20.2	109.267	11	<0.001 *
	Max	25.7	23.9	23.8	24.1	24.1	24.0	24.6	23.5	25.7	22.6	24.4	23.2	23.1			
	Median	22.1	21.7	21.9	23.5	22.8	21.9	21.6	21.8	22.8	21.6	22.4	21.3	21.8			
	Mean	22.2	22.2	22.2	23.3	23.0	21.3	22.0	21.9	23.0	21.7	22.4	21.3	21.9			
Level of relative humidity in (%)	Min	18.0	28.2	30.8	18.0	18.9	28.5	28.4	25.4	25.6	34.7	34.0	26.3	30.5	119.805	11	<0.001 *
	Max	49.9	38.9	37.3	35.1	37.0	49.9	42.0	39.2	42.0	41,5	42,1	48.0	44.2			
	Median	35.2	35.2	34.7	26.1	24.4	34.5	37.7	36.5	35.8	38.0	38.5	33.9	36.9			
	Mean	34.2	34.9	34.7	26.4	26.5	34.6	35.3	33.7	34.2	38.1	38.1	36.7	37.4			
Level of lighting in (lx)	Min	14.4	52.5	74.3	154.2	212.4	14.4	42.0	248.0	226.1	410.6	410.3	58.9	45.5	218.785	11	<0.001 *
	Max	869.1	384.7	290.1	702.1	734.2	686.1	869.1	396.2	401.5	569.5	574.2	257.2	188.9			
	Median	286.2	86.8	194.6	308.9	551.0	219.8	167.1	335.7	315.4	546.6	509.9	109.4	116.9			
	Mean	307.3	164.1	191.1	342.6	495.8	225.2	341.2	327.8	316.7	534.3	505.6	122.9	120.2			
Air quality in CO_2_ (ppm)	Min	456.0	614.0	863.0	474.0	472.0	615.0	689.0	527.0	456.0	509.0	505.0	737.0	989.0	246.942	11	<0.001 *
	Max	1352.0	1003.0	1103.0	830.0	1074.0	1174.0	1352.0	792.0	772.0	689.0	715.0	1221.0	1265.0			
	Median	751.5	854.0	999.5	556.0	672.0	847.5	889.0	648.0	656.5	619.0	625.0	958.0	1104.5			
	Mean	798.7	819.4	992.7	594.9	756.4	828.3	952.9	650.7	640.7	615.9	621.2	1004.9	1106.4			
Noise level in (dB(A))	Min	13.6	13.6	45.2	48.9	44.2	37.0	44.9	56.7	52.9	39.8	45.3	43.6	53.7	114.456	11	<0.001 *
	Max	86.3	76.1	86.3	74.3	75.1	70.2	76.2	70.9	76.4	62.9	70.6	76.2	79.3			
	Median	62.8	53.3	64.7	62.3	64.0	62.2	67.0	63.4	64.9	50.3	50.4	65.4	73.5			
	Mean	61.2	51.6	65.8	62.2	63.0	58.2	64.7	63.6	65.1	50.6	53.6	64.4	71.1			

Note: A, working counter of a nurses station at the surgical intensive care unit; B, working counter for preparation of medications at the surgical intensive care unit; C, working counter at the observation unit; D, overbed table as part of a hospital bed unit; E, working counter of an intervention area at the nursing ward; F, working counter in a nurses’ room at the nursing ward; G, working counter for the analysis of laboratory blood testing; H, working counter for the analysis of laboratory biochemical tests; I, working counter for the preparation of narcotic drugs; J, working counter for issuing pharmacy materials; K, working counter in the external admissions office for patients; L, working counter of the internal admissions office for patients; * *p* < 0.05 (Kruskal–Wallis test/Chi-square test); *n*, sample size; df, degrees of freedom.

**Table 4 ijerph-16-03224-t004:** Deviations (%) of obtained indoor environmental quality parameter values from the legally required and/or recommended values in the selected hospital environment.

Question and Deviations	Total (*n* = 360)	Total (*n* = 360)	Anaesthesiology and Reanimation Services		Emergency Department		Non-acute Physical Rehabilitation Services		Joint Medical Departments		Department for Medicinal Product Supplies		Surgical Admissions Unit		Chi-square	df	*p*
			A (*n* = 30)	B (*n* = 30)	C (*n* = 30)	D (*n* = 30)	E (*n* = 30)	F (*n* = 30)	G (*n* = 30)	H (*n* = 30)	I (*n* = 30)	J (*n* = 30)	K (*n* = 30)	L (*n* = 30)			
Do the obtained values deviate from the legally required and/or recommended values?	Temperature (% YES)	55.3	36.7	46.7	93.3	100.0	63.3	43.3	23.3	76.7	36.7	83.3	16.7	43.3	119.4	11.0	<0.001
	Relative humidity level (% YES)	21.7	13.3	0.0	66.7	66.7	3.3	33.3	33.3	33.3	0.0	0.0	10.0	0.0	133.5	11.0	<0.001
	Lighting level (% YES)	83.3	100.0	100.0	100.0	100.0	86.7	66.7	100.0	100.0	10.0	36.7	100.0	100.0	203.7	11.0	<0.001
	Air quality (% YES)	20.6	3.3	50.0	0.0	13.3	10.0	30.0	0.0	0.0	0.0	0.0	43.3	96.7	185.9	11.0	<0.001
	Noise level (% YES)	73.6	46.7	83.3	76.7	90.0	70.0	80.0	100.0	93.3	26.7	33.3	90.0	93.3	106.4	11.0	<0.001

Note: A, working counter of a nurses station at the surgical intensive care unit; B, working counter for preparation of medications at the surgical intensive care unit; C, working counter at the observation unit; D, overbed table as part of a hospital bed unit; E, working counter of an intervention area at the nursing ward; F, working counter in a nurses’ room at the nursing ward; G, working counter for the analysis of laboratory blood testing; H, working counter for the analysis of laboratory biochemical tests; I, working counter for the preparation of narcotic drugs; J, working counter for issuing pharmacy materials; K, working counter in the external admissions office for patients; L, working counter of the internal admissions office for patients; * *p* < 0.05; (Kruskal–Wallis test/Chi-square test); df, degrees of freedom; *n*, sample size.

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
