# Peer review of "Association between Sick Building Syndrome and Indoor Environmental Quality in Slovenian Hospitals: A Cross-Sectional Study"

_ijerph, 2019, doi:10.3390/ijerph16173224_

Round 1

Reviewer 1 Report

Dear authors,

the work focuses on an important issue, but I believe that quality of reporting (introduction and results) and conduction of the study (variables, statistics) could be improved.

I give you my suggestions to improve this paper:

Title  would sound better as: Association between Sick Building Syndrome and Indoor Environmental Quality in Slovenian hospitals: a cross-sectional study. Title should use key words, the study design and content of the research.

Starting from this title, you should improve the abstract, and the introduction. Indeed, in introduction (and title) arguing about health risk factors (biological, chemical etc) is useless, misleading and off-topic. The rationale of the study should be explained better. The only risk factor of which you should talk is the psychosocial one, very important in hospital, that you should consider by the MM040 that you have used to measure SBS. The objective should be revised based on my suggested title. So, in introduction you should argue only on SBS, IEQ, psychosocial risk in hospitals, studies on SBS, IEQ and psychosocial (if you have these data drawn by the MM040 questionnaire) in your country. 

Please, you should explain better in methods how you have used the MM040 questionnaire. I suggest you to dicotomize the results of MM040 questionnaire (see in literature) to obtain SBS yes and SBS not according to the number of symptoms. You should explain better the variables used (independent and outcome) and how you have treated (operationalize) the variables and statistics. I read in tables and results section some measures (SBS symptoms) that you should present before in methods section.

Results are a repetition of data included in tables. You should cut the result presentation and make an order: before descriptive statistics (mean and SD), after associations (better than correlation). You give too many details about field (not experimental) environmental measures.

I expected to find some important and well-known associations described in literature between environmental parameters and SBS symptoms. According to literature there is furthermore an association between SBS, psychosocial factors and IEQ parameters (noise, and others) and you should focus better and more on these relationships.

Good work!

Author Response

Dear Reviewer of the International Journal of Environmental Research and Public Health,

we have revised the manuscript according to the reviewers' comments (Manuscript ID: ijerph-578151; Type of manuscript: Article; Title Health Risk Factors for Sick Building Syndrome as a Trigger for Environmental Health Activities in Hospitals; Authors: Sedina Kalender Smajlović , Andreja Kukec , Mateja Dovjak *.

Our revisions are clearly highlighted, we used the "Track Changes" function in Microsoft Word. We explained point-by-point the details of the revisions in the manuscript and our responses to the reviewers' comments. We also included some rebuttals.

The authors would like to thank reviewers for their valuable time to comments / suggestions on the manuscript in a such constructive way.

Sincerely,

Sedina Kalender Smajlović

Andreja Kukec

Mateja Dovjak*

Reviewer 2 Report

Dear authors,

The study topic is interesting for health care professionals and the paper is well written. However, I have some recommendations to improve the paper.

The introduction provides a good background citing published papers related to the study topic. It gives a general idea about the study topic and the previous papers to the readers. I find it easy to understand. 

I think that the presentations of the study aims are a bit hard to understand because the sentence is too long. I would use shorter sentences to clarify the study aims. It might be good to use numbers to differentiate each aim.

I would include more information about the self assesment questionnaire. How many items it has? What are the possible responses to each item?

I would include an "ethical issues" section in the methods. 

How were the SBS symptoms and the other risk factors collected?

The data analysis must include the test used for the description of the sample, not only the chi square test.

I would include a table with the sample characteristic. It is very clear for the readers. 

In table 1 and 2 some numbers are in bold. Why?

In Table 1, in the rows that you say "f in %" I guess that the second number is the frequency. But I can not guess what is the first number (the number of people who have those number of symptoms?).

I would not use p= 0.000 in table 2 and 3. It is better to use <0.001

In the results sometimes you use the term "correlation" but you have not used that test. You have used a Chi square test. It might be better to find another term.

The conclusions of the study are very long (the first paragraph could be deleted because is not a response to the study aims.). The conclusions must answer the study aims in a clear and short way. 

The last part of the conclusion, starting with "these finding could be of great value..." I would move it to the discussion. 

Author Response

Dear Reviewer of the International Journal of Environmental Research and Public Health,

we have revised the manuscript according to the reviewers' comments (Manuscript ID: ijerph-578151; Type of manuscript: Article; Title Health Risk Factors for Sick Building Syndrome as a Trigger for Environmental Health Activities in Hospitals; Authors: Sedina Kalender Smajlović , Andreja Kukec , Mateja Dovjak *.

Our revisions are clearly highlighted, we used the "Track Changes" function in Microsoft Word. We explained point-by-point the details of the revisions in the manuscript and our responses to the reviewers' comments. We also included some rebuttals.

The authors would like to thank reviewers for their valuable time to comments / suggestions on the manuscript in a such constructive way.

Sincerely,

Sedina Kalender Smajlović

Andreja Kukec

Mateja Dovjak

Round 2

Reviewer 1 Report

Dear authors,

you have improved the paper that is very fine now.

However, I  still suggest you some minor improvements.

1) In abstract and main text I suggest you using "field measurements" instead of "experimental measurements".

2)In study's objective I suggest you to simplify, by reformulating as follows: ..1) estimate the prevalence of SBS symptoms...2)...to study the indoor air quality of selected hospital wards and compare them to ....3) to assess the association between SBS symptoms and indoor air quality.

3) In methods when you explain the methods used for IAQ measurements,please, you should indicate some references such as

Chirico F. Rulli G. Strategy and methods for the risk assessment of thermal comfort in the workplace. G Ital Med Lav Ergon. 2015 Oct-Dec;37(4):220-33.

Chirico F, Magnavita N. The significant role of health surveillance in the occupational heat stress assessment. Int J Biometerol. 2019;63(2):193-194. https://doi.org/10.1007/s00484-018-1651.

In methods or results, you should indicate the alpha of Cronbach of the MM040 (overall scale and subscales).

In methods of analysis I suggest to add to "parametric typical statistical values" "after checking for their normal distribution". But I wonder if only  "descriptive statistics" wouldn't better, because you are not sure before analysis of the normal distribution of all your parameters.

In discussion ("Inappropriate indoor environmental quality parameters"), please, you should indicate which legally or recommended values you  have followed (from EU? Slovenian laws? Technical standards?) You should indicate it also in methods section.

In the same paragraph of discussion, when you write about Italian researchers, you should cite one of the most important scholar on this issue, Prof Nicola Magnavita. Here, some his works on SBS in italian hospitals.

Magnavita N. Work-related symptoms in indoor environments: a puzzling problem for the occupational physician. Int Arch Occup Environ Health. 2015;88(2):185-196 10.1007/s00420-014-0952-7

Magnavita N. Psychosocial factors in indoor work-related symptoms. Application of the MM040/IAQ questionnaire. Med Lav 2014; 105 (4): 269-281

In limitations of the study, I suggest you to consider also the convenience sample (of hospital wards and healthcare workers) used (using a random sampling would be better) and the fact you have not considered psychosocial factors (there are some items about those in MM040, you should indicate this also in methods). Indeed, psychosocial factors could affect self-reported symptoms of SBS, and may be an important confounding factor.

In conclusions you indicate a "high prevalence of SBS symptoms", but in methods you have not categorized symptoms. You could define in methods categories (low, medium, high or low/high by dichotomizing on median value) or individuate two categories of SBS yes/not based on literature. Otherwise, I suggest to change the word "high"

Author Response

Subject: RESPONSE TO REVIEWER`S COMMENTS

Dear Editor and Reviewers of the International Journal of Environmental Research and Public Health,

we have revised the manuscript according to the reviewer`s comments, 2nd round (Manuscript ID: ijerph-578151); Type of manuscript: Article; Title Health Risk Factors for Sick Building Syndrome as a Trigger for Environmental Health Activities in Hospitals; Authors: Sedina Kalender Smajlović, Andreja Kukec , Mateja Dovjak *.

Our revisions (2nd round) are clearly highlighted, we used the "Track Changes" function in Microsoft Word. We explained point-by-point the details of the revisions in the manuscript and our responses to the reviewer`s comments. We also included some rebuttals.

The authors would like to thank the reviewer for his/her valuable time and all the comments / suggestions on the manuscript in such constructive way.

We strongly believe the contribution of this study warrants its publication in the Journal of Environmental Research and Public Health.
